# Diagnosing injection-production system faults in the same well using the rough set-LVQ neural network

Yi fang Yin👤*, Zunce Wang, Minzheng Jiang, Siyuan Chang

School of Mechanical Science and Engineering, Northeast Petroleum University, Daqing, PR China

* yyf@nepu.edu.cn

## Abstract

This study proposed a reverse calculation model of the unique rod pump injection and production system structures in the same well to diagnose and resolve defects, after which dynamometer diagrams of the system production and injection pumps were drawn. The invariant moment feature method was applied to identify seven such characteristics in the injection pump power graph, establishing a downhole system for fault diagnosis in rod pump injection and production systems in the same well using Rough Set(RS)-Learning Vector Quantization(LVQ). On the premise of keeping the classification ability unchanged, the Self-Organizing Map(SOM) neural network was used to discretize the original feature data, while RS theory was employed for attribute reduction. After establishing the LVQ fault diagnosis subsystem, the reduced decision table was entered for learning and training. The test results confirmed the efficacy and accuracy of this method in diagnosing downhole faults in rod pump injection-production systems in the same well. After comparing the test results with the actual working conditions, it can be seen that the rod pump injection-production diagnosis system based on RS-LVQ designed in this paper has a recognition rate of 91.3% for fault types, strong recognition ability, short diagnosis time, and A certain practicality. However, the research object of fault diagnosis in this paper is a single fault, and the actual downhole fault situation is complex, and there may be two or more fault types at the same time, which has certain limitations.

## 0. Introduction

Most oil fields in China have entered the high water cut stage. Excessive water cut increases production costs, and when production exceeds the economic limit, some oil wells are even forced to be abandoned [1, 2]. Therefore, same-well injection-production rod pumps effectively stabilize oil and water control in extra-high water-cut oil wells. Consequently, production and injection are combined in a single wellbore to successfully reduce the water content in the produced fluid [3]. However, the downhole section of this system consists of double injection and production pumps, presenting a more complicated structure than a conventional rod pump system. Failing to timeously identify and address issues adversely affects the production and economic value of oil fields, necessitating effective analysis of the system operational

**Funding:** This work is supported by the National Key Research and Development Program of China under grant (2022YFE0206700).

**Competing interests:** The authors have declared that no competing interests exist.

conditions, as well as accurate, reliable fault diagnosis and corresponding measure implementation [4, 5].

Since the 1960s, many studies have explored fault diagnosis techniques for pumping wells. Rapid scientific and technological development has prompted the emergence of new disciplines in various fields [6–9] and the progression of diagnostic pumping well approaches. Sun et al. [10] proposed the "five-finger test" analysis method relies on experienced workers to perceive the vibration of the polished rod with their hands, and then judge the working state of the underground according to their own experience. Although easy to implement, its accuracy was low, and it was gradually replaced by other methods. Li et al.[11] proposed the surface indicator diagram method mainly uses a dynamometer to measure the suspension point diagram and matches it with a standard dynamometer diagram to obtain the downhole working status. However, this method suffers from the following problems: there are too many assumed conditions, and some of the actual measured dynamometer diagrams are too singular in shape, and it is difficult to select the closest standard diagram, thus limiting its range of application. Qiu et al [12] proposed the Downhole dynamometer method using a downhole power instrument directly downhole to measure the dynamometer diagram of the oil pump. The method eliminates many of the uncertain factors and makes the test more accurate. However, all the downhole devices need to be taken out of the ground to install the downhole dynamic instrument, and the installation cost is steep, which is not conducive to practical promotion. Gibbs and Li [13, 14] proposed the Computer diagnosis method combines the mathematical model with the damping wave equation and the surface dynamometer diagram to obtain the corresponding downhole pump dynamometer diagram, and uses the dynamometer diagram to implement corresponding comparative analysis. Compared with the downhole dynamometer method, it not only saves the cost, but also improves the accuracy. However, this method still needs to rely on the skills and work experience of the operator, making it difficult to promote its use. Huang [15] proposed the "Artificial Intelligence Diagnosis" proposed by Using computer-aided diagnosis and intelligent techniques to analyze faults in the entire pumping unit, because of its advantages of simple operation and high precision, it has gradually become the current research field and has extensive development space. As shown in Table 1.

**Table 1. Fault diagnosis method.**

|  | Method | Advantages | Shortcomings |
|---|---|---|---|
| "Five Fingers Test" Analysis | Mainly rely on experienced staff to feel the vibration feeling of the polished rod in the palm, and use experience to infer the downhole working conditions | Simplicity of operator | Poor accuracy |
| Diagnosis method of ground dynamometer | Use the dynamometer to measure the suspension point dynamometer map, and then compare, match and interpret it with the standard map to infer the downhole working conditions | The type of failure can be more accurately defined | There are many assumptions, and the shape of some measured dynamometer diagrams is too singular to select the closest standard diagram, so its application range is limited. |
| Downhole dynamometer diagnosis method | Install the downhole power meter directly downhole to measure the dynamometer diagram of the oil well pump | Many uncertain factors can be eliminated, and the test accuracy is high | All downhole equipment needs to be driven out during installation, which is expensive and inconvenient for popularization and application |
| Computer diagnostics | Establish a mathematical model based on the damped wave equation, combine the system motion law derived from the surface dynamometer diagram, solve the downhole pump dynamometer diagram, and further apply the dynamometer diagram matching method to judge the downhole working conditions | Save costs, and improve Accuracy | It still needs to rely on the professional skills and work experience of the operator, and there is a certain difficulty in popularizing and applying it |
| AI diagnostics | Fault diagnosis of oil pumping system by using artificial intelligence method with the help of computer equipment | Easy to operate and high precision | The structural design of AI diagnosis is often determined by the designer based on experience, which has certain limitations. |

Usually, the working condition of the well can be preliminarily diagnosed through the suspended point indicator diagram, but if the working condition of the oil well is to be accurately diagnosed, it is best to use the pumping diagram. Since the downhole pump group of this system works at a depth of nearly one thousand meters underground, it is difficult to directly detect its working state parameters, and the method of using a downhole dynamic instrument is also infeasible due to factors such as inconvenient installation and high cost. A reasonable fault diagnosis model and drawing a pump power diagram are the core technologies for fault diagnosis. Through the theoretical calculation of the oil well production and the comparative analysis combined with the field survey, the statistics show that the probability of injection pump failure exceeds 90% when the injection-production system of an oil well fails, while the production pump can continue to operate normally. According to the commonly used method, using the suspension point dynamometer diagram for fault diagnosis, it is impossible to accurately identify and distinguish the fault types of the injection pump and the recovery pump, Therefore, based on an actual site scenario, this paper proposes a reverse calculation method that considers the production pump as a supplementary boundary condition and uses RS-LVQ to establish a downhole technique for diagnosing faults in the rod pump injection and production system in the same well. Therefore, defects are rapidly and accurately identified, allowing targeted measures to be implemented to address the issue [16–18], making up for the shortcomings of traditional methods.

# 1 The composition of the rod pump injection and production system in the same well

The rod pump injection and production system in the same well mainly consist of a conventional beam pumping unit, a production pump, a sealing piston, a bridge packer, an injection pump, and an oil-water separation system [3], as shown in Fig 1.

The oil enters the settling cup of the oil-water separator, followed by the production fluid in the oil layer, and is rapidly separated into a denser water layer and a less dense low-water oil layer due to gravity. The water in the lower layer flows through the center pipe of the oil-water separation device to the suction port of the injection pump, where it is circulated and re-injected into the injection layer. The concentrated oil in the upper layer flows up to the suction port of the production pump through the central flow channel and is lifted to the ground.

# 2 Diagnostic model establishment and numerical analysis

## 2.1 Diagnostic model establishment

The longitudinal vibration equation of the sucker rod string in the rod pump injection and production system in the same well [19] is:

$$E_r A_r \frac{\partial^2 u}{\partial x^2} = \rho_r A_r \frac{\partial^2 u}{\partial t^2} - \rho_r A_r g + v_e \frac{\partial u}{\partial t} \tag{1}$$

where $E_r$ is the elasticity modulus of the sucker rod, Pa; $A_r$ is the sucker rod cross-sectional area, m$^2$; $\rho_r$ is the sucker rod density, kg/m$^2$; and $v_e$ is the drag coefficient per unit length.

The diagnostic model of the rod pump injection-production system in the same well can be summarized as the initial boundary value problem of the wave equation as follows:

$$c^2 \frac{\partial^2 u}{\partial t^2} = \frac{\partial^2 u}{\partial t^2} + v \frac{\partial u}{\partial t}, 0 < x < L, 0 < t \leq T \tag{2}$$

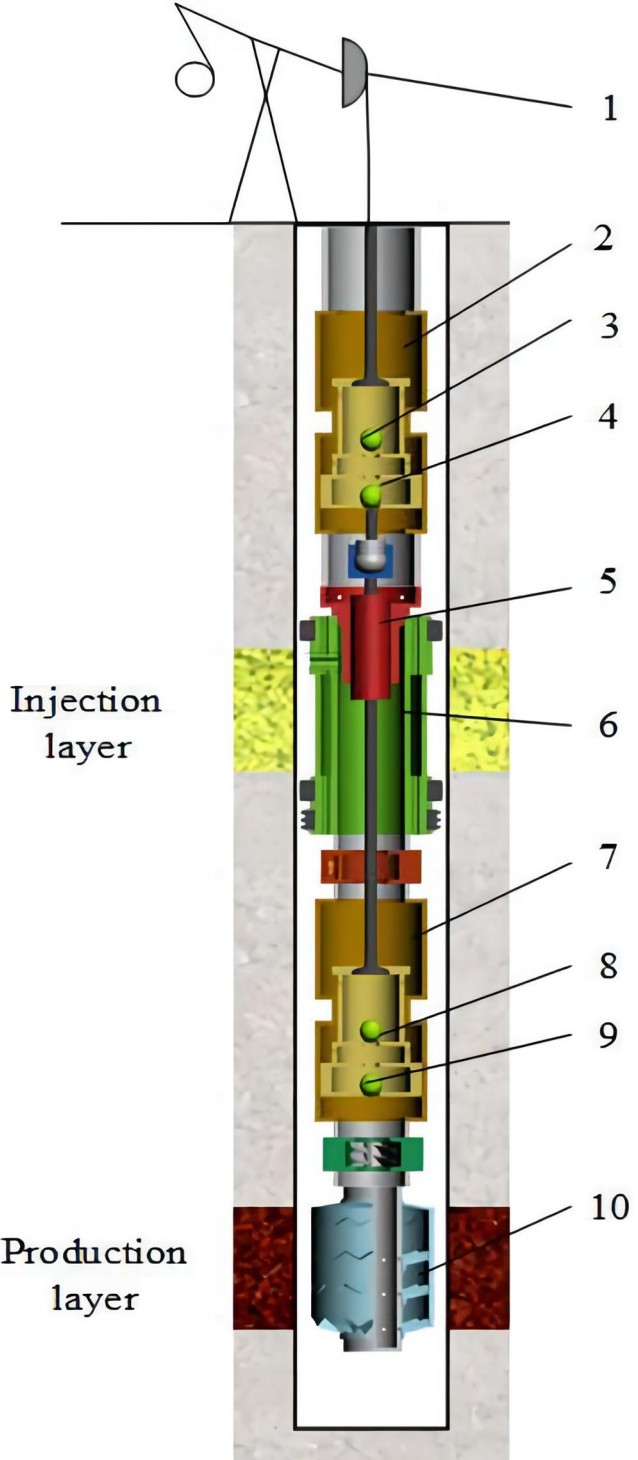

**Fig 1. A schematic diagram of the rod pump injection and production system in the same well.** 1. The conventional beam pumping unit. 2. The production pump. 3. The upper production pump valve. 4. The lower production pump valve. 5. The sealing piston. 6. The bridge packer. 7. The injection production pump. 8. The upper injection pump valve. 9. The lower injection pump valve. 10. The oil-water separator.

where:

$$c = \sqrt{\frac{E_r}{\rho_r}}, v = \frac{v_e}{\rho_r A_r}$$

The initial condition is:

$$u|_{t=0} = \phi(x), \frac{\partial u}{\partial t}\bigg|_{t=0} = \psi(x), 0 \le x \le L \tag{3}$$

The boundary condition is:

$$u|_{x=0} = \boldsymbol{\Phi}_1(t), u|_{x=L_1} = \boldsymbol{\Phi}_2(t), u|_{x=L} = \boldsymbol{\Phi}_3(t), 0 \le t \le T \tag{4}$$

where $c$ is the transmission speed of the sound wave in the pole column, m/s; $L_1$ is the length of the upper sucker rod in the production pump, m; $L_2$ is the length of the upper sucker rod in the injection pump, m; $L$ is the total length of the sucker rod, m; $u$ is the elastic displacement of any sucker rod section, m; $x$ is the distance from the suspension point to any sucker rod section, m; $t$ is the time from the starting point, s; $v$ is the damping coefficient of the well fluid on the sucker rod; and $T$ is the cycle. $\phi(x)$, $\psi(x)$, $\Phi_1(t)$, $\Phi_2(t)$, and $\Phi_3(t)$ are determined by the working conditions of the pumping system.

Establish and solve the static/dynamic load model of the production pump, and obtain the load-displacement function of the production pump. The load-displacement function of the suspension point derived from the suspension point indicator diagram is subtracted from the calculated load-displacement function of the production pump to obtain the load-displacement function of the injection pump so that the working condition diagnosis can be performed according to the injection pump work diagram.

## 2.2 Reverse calculation method

The formula above shows that three boundary conditions must be known when solving the diagnostic model. The displacement and load boundary conditions of $x = 0$ can be obtained from the suspension point dynamometer diagram, but the boundary conditions of $x = L_1$ are missing. However, it is difficult to directly detect the dynamometer diagram of the downhole pump with existing technology. Field investigation statistics show that the failure of the injection pump accounts for more than 90% of the total failure, while the production pump works normally. Therefore, a reverse calculation method is proposed, assuming normal production pump operation. This technique is combined with the display difference and iterative methods to solve the fault diagnosis model of the rod pump injection-production system in the same well. The injection pump dynamometer is obtained by solving and drawing the production pump indicator diagram and combining it with the measured dynamometer diagram of the suspension point on the ground.

The reverse calculation method is based on the assumption that the production pump is working normally, the fault diagnosis model of the injection-production system of the rod pump in the same well is solved by the complement grid method to obtain the production pump power diagram, and then the injection pump power diagram is obtained by further solving the collected suspension point diagram. In the normal working state of the production pump, the solution steps of the fault diagnosis model as shown in Fig 2.

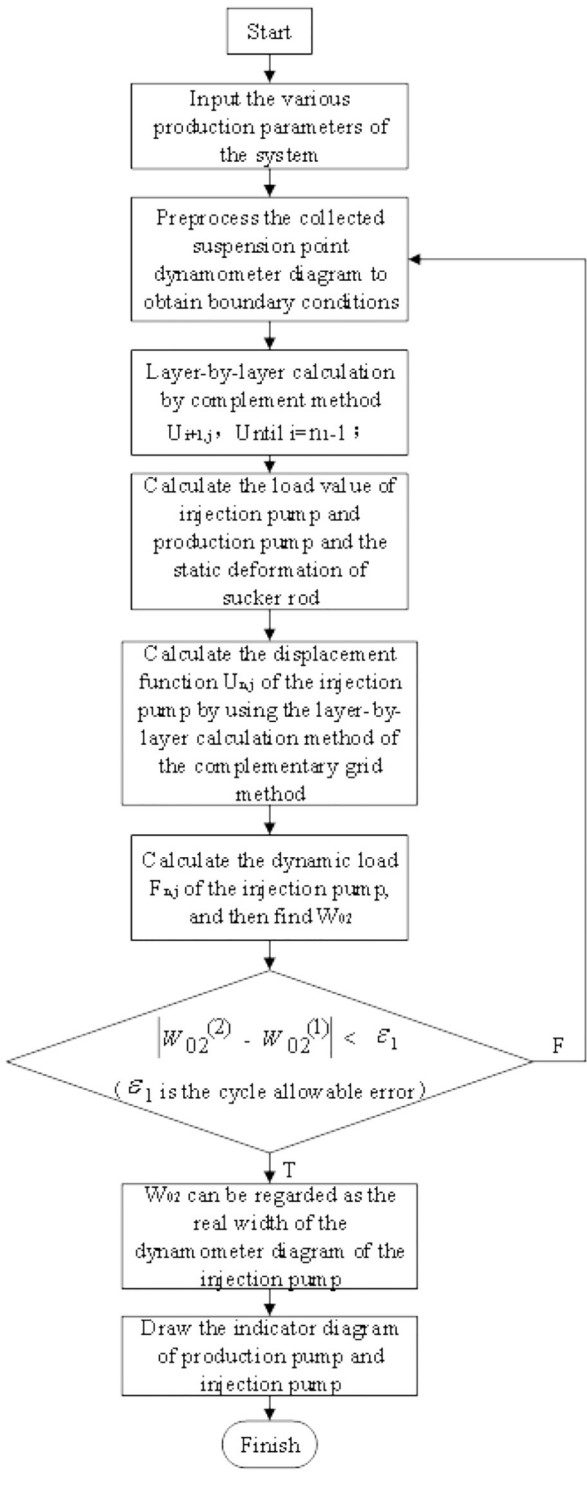

**Fig 2. The solution steps of the fault diagnosis model.**

## 2.3 Numerical analysis

Solving the model was divided into three sections: the upper sucker rod of the production pump, the upper sucker rod of the injection pump, and the interface. The solution area was divided into several rectangular grids, after which steps $h_1$ and $h_2$ in direction x and steps $\tau$ and $\tau = \frac{T}{m}$ in direction t are used to determine the grid coordinates as follows:

$$x = ih_1, i = 0, 1, 2, \cdots, n_1 \tag{5}$$

$$x = L_1 + (i - n_1)h_2, i = n_1 + 1, \cdots, n \tag{6}$$

$$t = j\tau, j = 0, 1, 2, \cdots, m \tag{7}$$

where $h_1$ and $h_2$, $\tau$ represent constants exceeding zero, $n = n_1 + n_2$.

The upper sucker rod diagnostic models of the production and injection pumps involved the problem of the same rod diameter. The different formats were found in the available literature [20], while the interface was deduced using continuous conditional Formulas (8) and (9).

$$(F_{i,j})_1 = (F_{i,j})_2 + F_j \tag{8}$$

$$(u_{i,j})_1 = (u_{i,j})_2 \tag{9}$$

where $(F_{i,j})_1$, $(F_{i,j})_2$ are the loads at the interface of the upper sucker rods of the production and injection pumps, respectively, N; and $F_j$ is the production pump load, N.

Formula (8) can also be expressed as:

$$E_r A_r \left(\frac{\partial u}{\partial x}\right)_{i1} = E_r A_r \left(\frac{\partial u}{\partial x}\right)_{i2} + F_j \tag{10}$$

A method obtained from the available literature [20] was used to deduce Formulas (9) and (10) as follows:

$$\begin{aligned} \alpha_1(u_{i,j+1} - 2u_{i,j} + u_{i,j-1}) + \beta_1(u_{i,j+1} - u_{i,j}) + \gamma_1(u_{i,j} + u_{i-1,j}) \\ = -\alpha_2(u_{i,j+1} - 2u_{i,j} + u_{i,j-1}) + \beta_2(u_{i,j+1} - u_{i,j}) + \gamma_2(u_{i,j} + u_{i-1,j}) + F_j \end{aligned} \tag{11}$$

where $\alpha = \frac{hE_r A_r}{2(c\tau)^2}$, $\beta = \frac{hE_r A_r v}{2c\tau}$, $\gamma = \frac{E_r A_r}{2\tau}$, and subscripts 1 and 2 represent the upper sucker rods of the production and injection pumps, respectively.

The differential equation is expressed as:

$$u_{i+1,j} = \frac{1}{\gamma_2} \begin{bmatrix} (\alpha_s + \beta_s)u_{i,j+1} + u_{i,j-1}\alpha_s \\ -u_{i-1,j}\gamma_1 - u_{i,j}(2\alpha_s + \beta_s) \\ -\gamma_1 - \gamma_2) - F_j \end{bmatrix} \tag{12}$$

where $\alpha_s = \alpha_1 + \alpha_2$, $\beta_s = \beta_1 + \beta_2$.

The dynamic load at the interface is:

$$(F_{i,j})_1 = \alpha_1(u_{i,j+1} - 2u_{i,j} + u_{i,j-1}) + \beta_1(u_{i,j+1} - u_{i,j}) + \gamma_1(u_{i,j} + u_{i-1,j}) \tag{13}$$

$$(F_{i,j})_2 = -\alpha_2(u_{i,j+1} - 2u_{i,j} + u_{i,j-1}) - \beta_2(u_{i,j+1} - u_{i,j}) + \gamma_2(u_{i,j} + u_{i-1,j}) \tag{14}$$

$$F_j = (F_{i,j})_1 - (F_{i,j})_2 \tag{15}$$

## 3 Data acquisition and preprocessing

### 3.1 Data acquisition

After solving the diagnostic model of the injection-production system of the rod pump in the same well, the dynamometer diagram of the injection pump was drawn and combined with the actual working conditions on site and summarized as upper valve leakage. The eight fault types include rod breakage, insufficient liquid supply, continuous spraying with pumping, downward impact on the pump, gas impact, lower valve leakage, and simultaneous upper and lower valve leakage. The corresponding fault types are shown in Fig 3. Furthermore, 770 injection pump indicator diagrams with relatively distinctive features were selected to establish a sample library, as shown in Table 2.

### 3.2 Data preprocessing

**3.2.1 The coordinate digitization of the injection pump dynamometer diagram.** To facilitate the possibility of image feature extraction, obtaining the pixel coordinates of the curve outline is necessary. As shown in Fig 4, the Moore neighborhood boundary tracking algorithm was used to randomly select an initial point $P_0$ on the curve, search the eight neighborhoods around this point, and select the corresponding point with the smallest difference between the pixel value and $P_0$, which is denoted as $P_1$. This process was repeated until returning to the initial point, $P_0 = P_N$ while storing each curve pixel point, $P_0, P_1 \ldots, P_N$.

**3.2.2 Normalization of the injection pump dynamometer diagram.** The sizes of the injection pump dynamometer diagrams drawn according to the calculation differed. The sizes and number of points were normalized to compare the feature value extraction.

**1. Size normalization**

The mapminmax linear function conversion method was used to normalize the size of the data row by row according to the following calculation formula:

$$x = ih_1, X = 1 - (x - X_{\min})/(X_{\max} - X_{\min})$$

$$Y = (y - Y_{\min})/(Y_{\max} - Y_{\min})$$

When the data of an entire row were equal, that is, $X_{\max} = X_{\min}$, the data processing software adjusted this transformation to $y = Y_{\min}$ by default. Therefore, both the $X_{\max}$ and $Y_{\max}$ normalized point coordinates were equal to 1.

**2. Point normalization**

The curve points in the injection pump dynamometer diagram were normalized to 300 to obtain the transformed coordinates of each point. Fig 5 shows the step flow of this algorithm, while Fig 6 presents the power display image of an injection pump after point normalization.

**3.2.3 Binarization of the injection pump dynamometer diagram.** A binary image refers to an image in which the pixels are either black or white (gray value = 0 or 1), with no

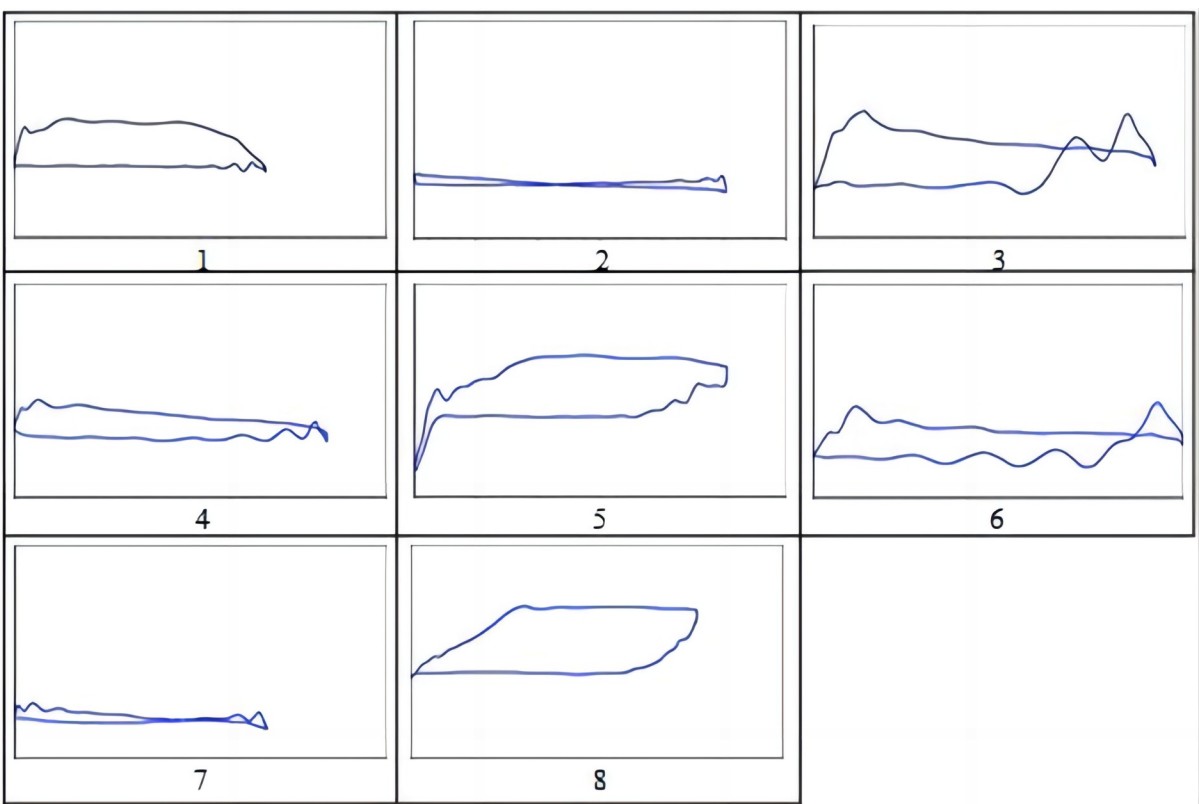

**Fig 3. Injection pump failure.**

**Table 2. Number of sample banks.**

| Sample No. | Fault type | Number of samples | Sample No. | Fault type | Number of samples |
|---|---|---|---|---|---|
| 1 | Upper valve leakage | 80 | 5 | Bump pump | 136 |
| 2 | Sucker rod fracture | 103 | 6 | Air influence | 80 |
| 3 | Insufficient fluid supply | 140 | 7 | Lower valve leakage | 35 |
| 4 | Even spray and pump | 160 | 8 | Simultaneous upper and lower valve leakage | 36 |

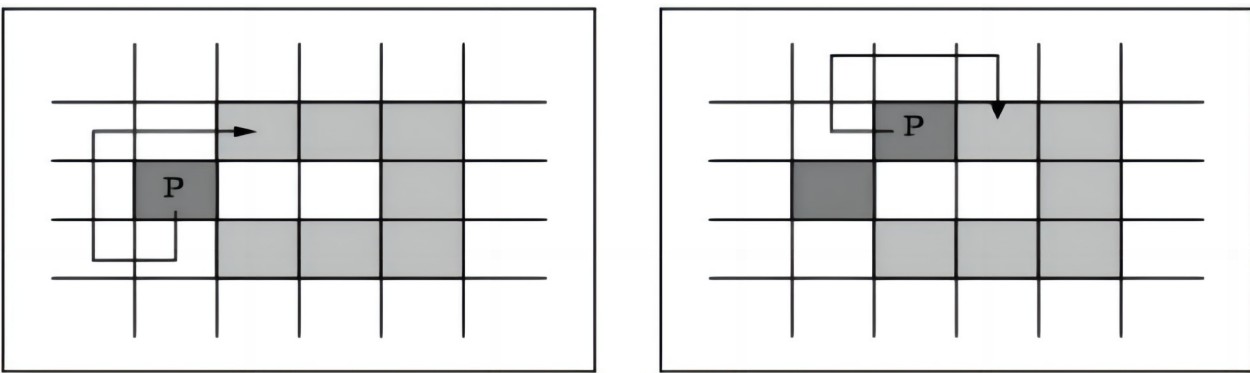

**Fig 4. A schematic diagram of the tracking process of Moore's eight-neighborhood boundary tracking algorithm.**

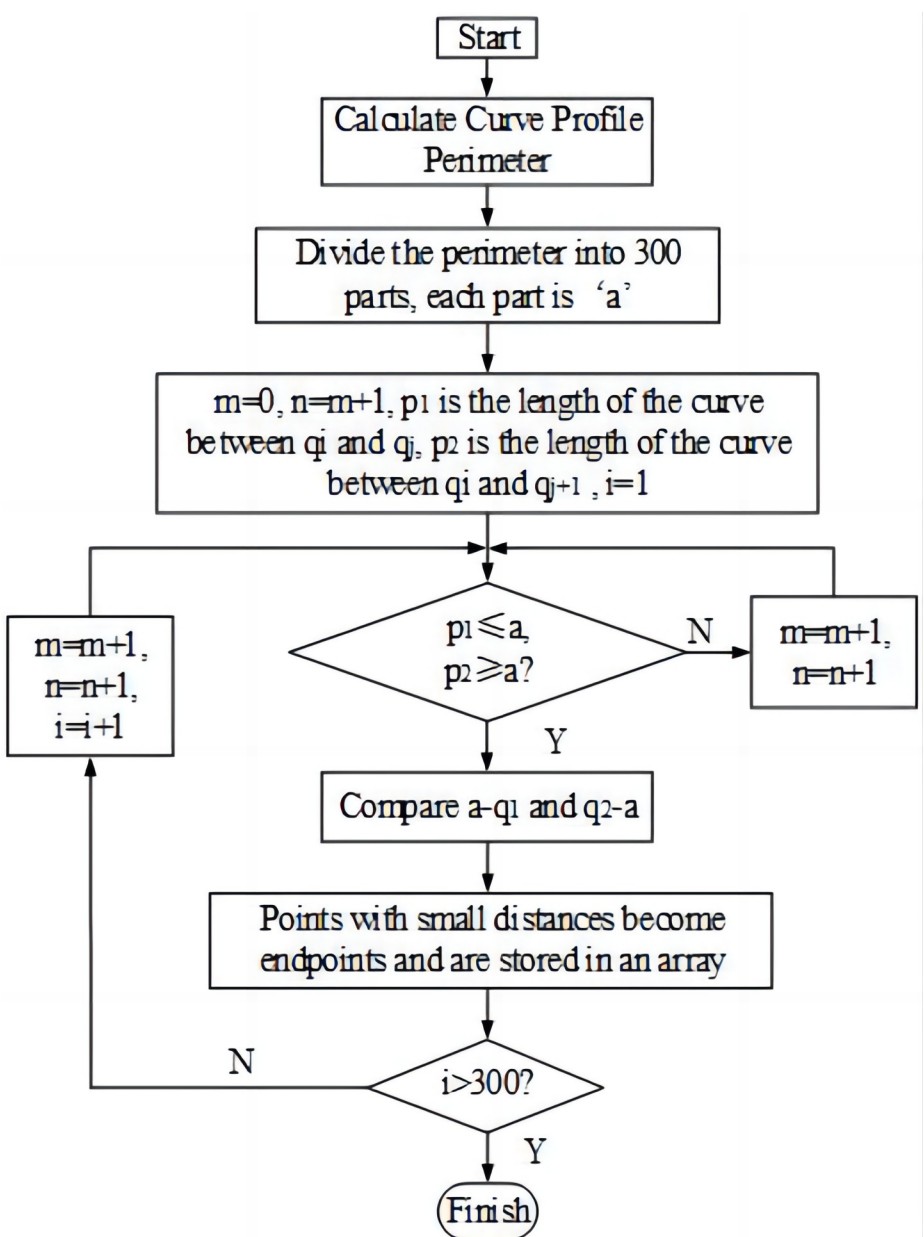

**Fig 5. A flowchart of the dynamometer point normalization.**

intermediate gray value transition. This can simplify the image, significantly reduce the amount of data, and highlight the outline of a target object in good condition. It is also convenient for further image processing.

Therefore, to ensure the accuracy of the subsequent eigenvalue extraction results, the MATLAB data processing software was used for image grayscale binarization, during which any threshold T and the original image f(x,y) at each point were selected. The gray and T values were individually compared. If $f(x, y) \geq T$, the gray value of this point was set to 1, otherwise it was set to 0. Then, the coordinates of each curve point were determined via $[x, y] = find$ $(BW = 1)$, and stored in the corresponding matrix. Fig 7 shows the dynamometer diagram before and after binarization.

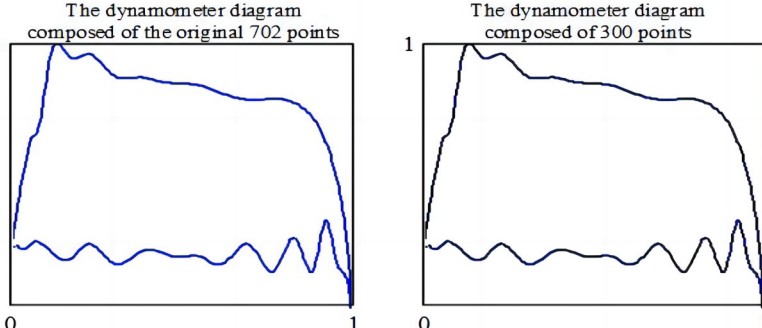

**Fig 6. The indicator diagram after point normalization.**

## 4 Feature extraction of injection pump dynamometer data

### 4.1 Hu moment invariant characteristics

The key to injection pump power map recognition is extracting the representative eigenvalues. The invariant moment method can describe the global characteristics of the image shape [21, 22]. The method is unique, that is, for an image with limited area and continuous piecewise density distribution function, each order moment exists, and its set is the only description of all information of the image. The gray pixel value of a binary image can be represented by a $N_1 \times N_2$ matrix f(x,y), as shown in Fig 8.

Then, the two-dimensional (p+q) moment of the region f(x,y) is defined as:

$$m_{pq} = \sum_x \sum_y x^p x^q f(x,y), p, q = 0, 1 \ldots \qquad (16)$$

The corresponding center distance is defined as:

$$u_{pq} = \sum_x \sum_y (x - \bar{x})^p (y - \bar{y})^q f(x,y), p, q = 0, 1, 2 \ldots \qquad (17)$$

where $\bar{x} = \frac{m_{10}}{m_{00}}, \bar{y} = \frac{m_{01}}{m_{00}}, (\bar{x}, \bar{y})$ is the gray centroid of the target area.

The normalized (p+q) order central moment of f(x,y) is defined as:

$$\eta_{pq} = \frac{u_{pq}}{u_{00}^\gamma}, p, q = 0, 1, 2 \ldots \qquad (18)$$

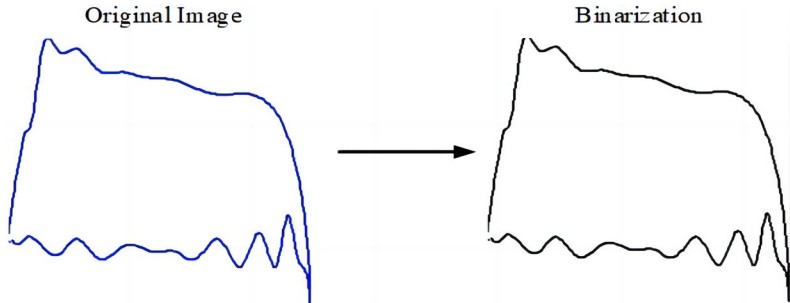

**Fig 7. The binary dynamometer diagram.**

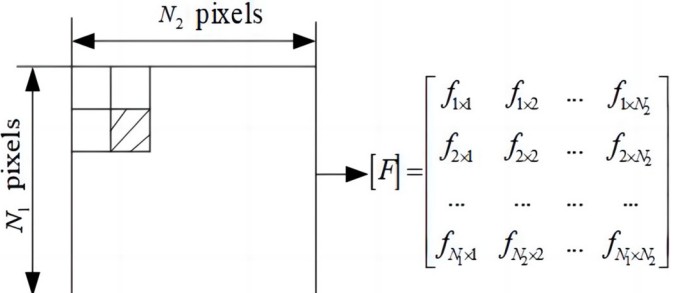

**Fig 8. The matrix representation of the digital image.**

where

$$\gamma = \frac{p+q}{2} + 1, p, q = 2, 3 \tag{19}$$

After algebraic transformation, the following seven two-dimensional invariant Hu moments were obtained:

$$\varphi_1 = \eta_{20} + \eta_{02} \tag{20}$$

$$\varphi_2 = (\eta_{20} - \eta_{02})^2 + 4\eta_{11}^2 \tag{21}$$

$$\varphi_3 = (\eta_{30} - 3\eta_{12})^2 + (3\eta_{21} - \eta_{03})^2 \tag{22}$$

$$\varphi_4 = (\eta_{30} + \eta_{12})^2 + (\eta_{21} + \eta_{03})^2 \tag{23}$$

$$\varphi_5 = (\eta_{30} - 3\eta_{12})(\eta_{30} + \eta_{12}) \begin{bmatrix} (\eta_{30} + \eta_{12})^2 \\ -3(\eta_{21} + \eta_{03})^2 \end{bmatrix} \\ +(3\eta_{12} - \eta_{03})(\eta_{21} + \eta_{30}) \begin{bmatrix} 3(\eta_{30} + \eta_{12})^2 \\ -(\eta_{21} + \eta_{03})^2 \end{bmatrix} \tag{24}$$

$$\varphi_6 = (\eta_{20} - \eta_{02}) \begin{bmatrix} (\eta_{30} + \eta_{12})^2 \\ -(\eta_{21} + \eta_{03})^2 \end{bmatrix} \\ +4\eta_{11}(\eta_{30} + \eta_{12})(\eta_{21} + \eta_{03}) \tag{25}$$

$$\varphi_7 = (3\eta_{21} - \eta_{02})(\eta_{30} + \eta_{12}) \\ \left[ (\eta_{30} + \eta_{12})^2 - 3(\eta_{21} + \eta_{03})^2 \right] \\ +(3\eta_{12} - \eta_{03})(\eta_{21} + \eta_{30}) \\ \left[ 3(\eta_{30} + \eta_{12})^2 - (\eta_{21} + \eta_{03})^2 \right] \tag{26}$$

**Table 3. The eigenvalues of the Hu moment invariants for eight fault types.**

| Moment invariant / Fault | $\varphi_1$ | $\varphi_2$ | $\varphi_3$ | $\varphi_4$ | $\varphi_5$ | $\varphi_6$ | $\varphi_7$ |
|---|---|---|---|---|---|---|---|
| Upper valve leakage | 0.2385409 | 1.6869326 | 3.7437504 | 3.9289875 | 8.0979815 | 5.3120067 | 7.9797591 |
| Sucker rod fracture | 0.1101338 | 0.8635656 | 2.2821173 | 3.5895577 | 10.305678 | 4.5154081 | 7.5979941 |
| Insufficient fluid supply | 0.0112114 | 3.4366969 | 2.3119035 | 6.9908198 | 11.786280 | 9.5791496 | 12.782736 |
| Even spray and pump | 0.3234390 | 2.3343733 | 0.7474746 | 5.7835012 | 10.586089 | 8.1306606 | 9.6650600 |
| Bump pump | 0.3943769 | 2.2091004 | 2.3314828 | 5.3796299 | 12.041580 | 6.6995961 | 9.2448878 |
| Air influence | 0.5004917 | 4.6420833 | 1.7663979 | 2.9748834 | 7.4212813 | 6.1598422 | 5.3670010 |
| Lower valve leakage | 0.1239436 | 1.3272180 | 3.0636480 | 6.4556739 | 11.837090 | 9.4220433 | 11.747093 |
| Simultaneous upper and lower valve leakage | 0.3348788 | 1.2840949 | 2.2297673 | 6.2479491 | 14.740239 | 7.6623900 | 10.522609 |

The seven Hu moment invariant features of the binarized injection pump dynamometer diagram were extracted using MATLAB software and stored in a feature vector $\varphi = [\varphi_1, \varphi_2, \varphi_3, \varphi_4, \varphi_5, \varphi_6, \varphi_7]$ as its identification feature. Some data are listed in Table 3.

## 4.2 Morphological eigenvalues

Morphological characteristics represent key features of the injection pump dynamometer curve and are vital for image recognition. Therefore, to normalize and binarize the injection pump dynamometer diagram, this section established a relationship between an image of known size and a pixel and extracted four parameters, including area, rectangularity, elongation, and Euler number. The morphological feature values of the images in the sample library were extracted using MATLAB software, providing basic data for subsequent diagnosis and identification, as listed in Table 4.

# 5 The RS-LVQ downhole fault diagnosis system for injection and production rod pumps

## 5.1 The RS-LVQ model

The rough set-LVQ fault diagnosis model reflected the direct interconnected relationship between the core attribute in the fault feature and the output neuron of the LVQ neural network. The training sample data were analyzed and organized, after which an initial decision table was created, and the continuous attribute was discretized using the SOM neural network. Then, RS theory was employed to reduce the conditional attributes in the discrete data table, obtain the minimum conditional attribute set and kernel, denoting the final decision table,

**Table 4. The morphological eigenvalues of eight fault types.**

| Fault | Area | Rectangularity | Elongation | Euler number |
|---|---|---|---|---|
| Upper valve leakage | 13463.25 | 0.172360487 | 0.885521886 | -227 |
| Sucker rod fracture | 12991.625 | 0.17638244 | 0.835016835 | -172 |
| Insufficient fluid supply | 17629.75 | 0.213523121 | 0.936026936 | -309 |
| Even spray and pump | 14430.25 | 0.171684453 | 0.952861953 | -235 |
| Bump pump | 11983.5 | 0.139614134 | 0.973063973 | -155 |
| Air influence | 12903.5 | 0.151380237 | 0.966329966 | -189 |
| Lower valve leakage | 13300.375 | 0.157684533 | 0.956228956 | -228 |
| Simultaneous upper and lower valve leakage | 13912.25 | 0.167894597 | 0.939393939 | -240 |

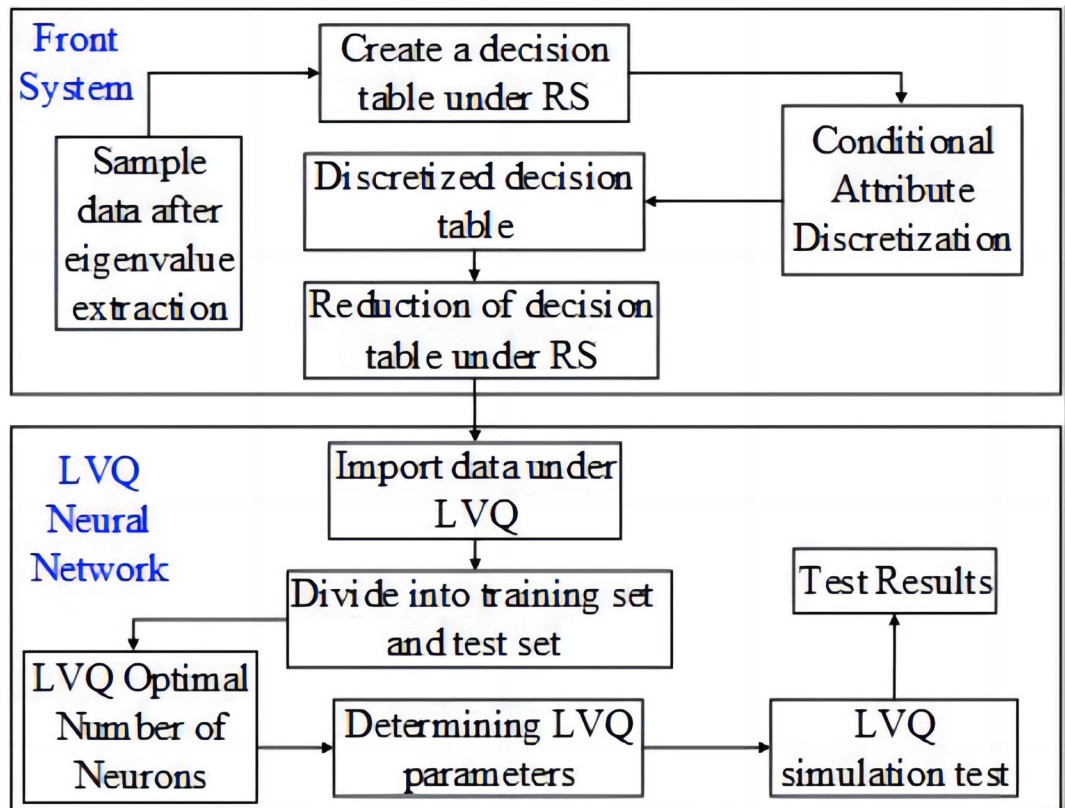

**Fig 9. The RS-LVQ neural network fault diagnosis system structural schematic.**

and determine the optimal neural network structure. The K-level cross-validation method was used to determine the optimal number of neurons. The reduced data were entered as samples for learning and training. After stabilizing the LVQ network test, the extracted eigenvalues of the injection pump dynamometer diagram were entered into the network for cluster analysis after discretization to obtain the fault results.

As shown in Fig 9, the structure-loose coupling method was adopted, while the rough set was used as the front-end system for data preprocessing. The final reduced decision table data was imported into the LVQ system for fault diagnosis.

## 5.2 Developing the RS-LVQ diagnostic system

**5.2.1 Establishing the decision table.** The extracted feature data was used to establish a decision table with the following conditional attributes: $C_1$-$C_7$ represented the seven Hu moment invariant features of $\varphi_1$-$\varphi_7$, while $C_8$-$C_{11}$ denoted the area, rectangularity, elongation, and Euler number, respectively. D represented the decision attribute, while 1–8 in the first column denoted the upper valve leakage, broken rod, insufficient liquid supply, continuous spraying and pumping, lower impact pump, air influence, lower valve leakage, and simultaneous upper and lower valve leakage A total in the eight failure conditions. The MATLAB software was employed to extract the data in the decision table from the eigenvalues of the dynamometer diagram of the injection pump in the rod pump injection-production system and shown in the table as part of the original data decision Table 5.

**Table 5. The partial raw data decision table.**

|  | $C_1$ | $C_2$ | $C_3$ | $C_4$ | $C_5$ | | |
|---|---|---|---|---|---|---|---|
| 1 | 0.25436086 | 2.07229706 | 3.17274798 | 3.50093951 | 8.52988171 | | |
| 2 | 0.16307278 | 1.31189238 | 2.70071645 | 3.86086367 | 10.0970147 | | |
| 3 | 0.16380084 | 3.67749806 | 1.84545060 | 7.02102655 | 11.8872480 | | |
| 4 | 0.31897295 | 2.54441441 | 0.94357150 | 3.17872211 | 6.44792400 | | |
| 5 | 0.38258014 | 1.49489412 | 0.16161059 | 2.50513097 | 4.97429860 | | |
| 6 | 0.00433893 | 2.99945057 | 2.01117414 | 4.54933023 | 8.64833761 | | |
| 7 | 0.17726165 | 0.98213087 | 1.37201278 | 3.94148253 | 7.47841100 | | |
| 8 | 0.42793307 | 1.71081437 | 3.39996392 | 4.35965522 | 11.0077026 | | |
| $C_6$ | $C_7$ | $C_8$ | $C_9$ | $C_{10}$ | $C_{11}$ | D |
| 4.61853943 | 7.09482179 | 13478.625 | 0.15979780 | 0.95622895 | -253 | 1 |
| 8.99275436 | 7.81207074 | 12991.625 | 0.17638244 | 0.83501683 | -172 | 2 |
| 9.65368996 | 13.7529888 | 17629.75 | 0.21352312 | 0.93602693 | -309 | 3 |
| 4.64316284 | 5.43458609 | 12565.375 | 0.15440751 | 0.92255892 | -196 | 4 |
| 4.78630321 | 6.23869147 | 10952 | 0.13708334 | 0.90572390 | -166 | 5 |
| 6.76399449 | 8.67941947 | 16146.125 | 0.19209914 | 0.95286195 | -297 | 6 |
| 6.90119722 | 7.45720611 | 13300.375 | 0.15768453 | 0.95622895 | -228 | 7 |
| 5.28616333 | 8.25755204 | 11654.5 | 0.13347190 | 0.98989899 | -170 | 8 |

**5.2.2 Discretization of the continuous attributes.** The analysis and processing capability of the rough set theory only extends to symbolic attribute value discretization. However, the eigenvalues extracted in Section 4 represent continuous attribute values. Therefore, this data must be discretely processed before using the rough set theory to reduce the fault decision table and extract the rules.

**1) Algorithm steps**

This paper used the dynamic SOM neural network method to discretize the continuous data attributes according to the following algorithm steps:

1. Raw data table input: $DT = (U, C, D, V, f)$. Establishing the continuous properties: $c_j(j = 1, 2, \cdots, m)$.

2. Initialization: $\alpha_{max} \Leftarrow \varepsilon, r \Leftarrow r_{min}$. The network structure was established as $[1, r]$.

3. The $SOM_j$ clusters for $c_j$ was established to obtain $c_j^r, j = 1, 2, \cdots, m$ and calculate the $\alpha$ values.

4. If $\alpha \leq \alpha_{max}$, skip to 9);
   Otherwise, Equation $v' = (v - min_c)(max_{new} - min_{new})/(max_c - min_c) + min_{new}$ was adopted to eliminate the dimension effect and normalize the original C set.

5. The attribute variance was calculated to obtain $C_0 = \left\{ c_{0_1}, c_{0_2} \cdots c_{0_m} \right\}$ (arranged from small to large).

6. $j \Leftarrow 1, r \Leftarrow r + 1$.

7. $c_{0_1}$ was clustered on the attribute to obtain $c_{0_j}^r$, and the $\alpha$ value was calculated.

8. If $\alpha \leq \alpha_{max}$, the process continued to 9). Otherwise, if $j < m$, then $j \Leftarrow j + 1$, skip to 7).for calculation. If $j = m$, then $j \Leftarrow 1, r \Leftarrow r + 1$, and the process continued to 7) for calculation.

**Table 6. Parameters of network training.**

|  | Input layer | Hidden layers | Output layer |
|---|---|---|---|
| Number of neurons | 5 | 13 | 8 |
| Number of Training Epochs |  | 100 |  |
| Performance Target |  | 0.005 |  |
| Number of Validation Checks |  | 1000 |  |
| Learning Rate |  | 0.5 |  |

9. Output $DT^p$, where $\alpha$ denoted incompatibility, $\alpha = 1 - \gamma_C(D)$, $\alpha_{\max}$ was the maximum allowable value of incompatibility, $\varepsilon$ signified the constant coefficient (smaller non-negative number), $r_{\min}$ represented the initial cluster number, $c_j^r$ denoted the discrete attribute column, and $DT^p$ signified the discretization decision table.

The meanings of other symbols were the same as those shown above.

**2) Algorithm implementation**

Table 4 shows eight sample vector sets, with each sample containing 11 conditional attributes, which were simulated, studied, and trained. The running effect of the program is shown in Table 6. The network topology and distance between the adjacent neurons are illustrated in Figs 10 and 11, respectively. The clustering results are presented in Table 7.

The analysis of the above table yielded the following conclusions:

1. When the number of network training steps was set to 10, the clustering results of samples 1, 2, 4, 5, 7, and 8 were the same, while those of samples 3 and 6 varied. The samples were preliminarily classified.

2. When the number of network training steps was set to 50, the clustering results were the same between samples 1 and 7, 3 and 6, and 5 and 8, while samples 2 and 4 were equal. However, the classification was not sufficient.

3. When the number of network training steps was set to 60 or even 80, it was impossible to completely distinguish all the samples. When the number of training steps reached 100, the clustering results showed differentiation between all samples. The expected effect was

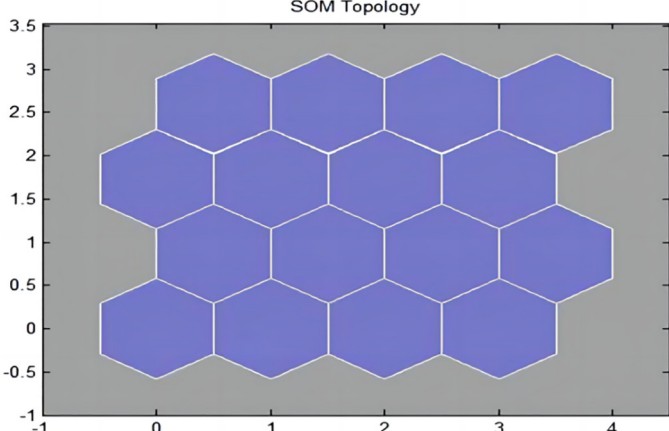

**Fig 10. A schematic diagram of the network topology structure.**

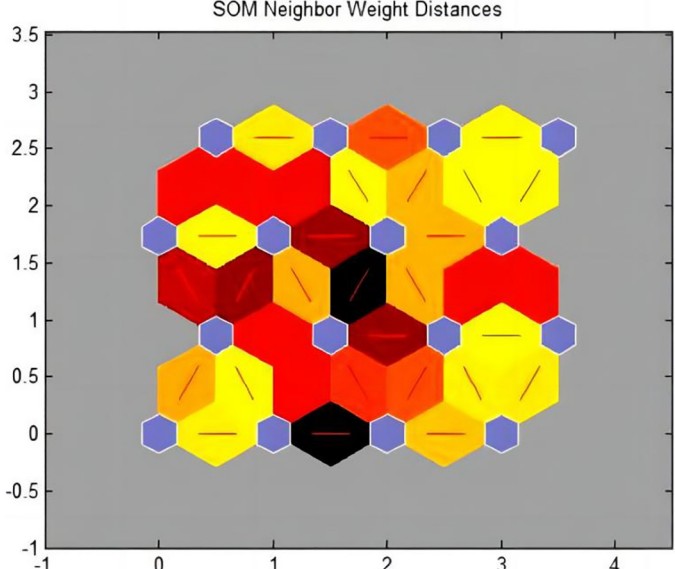

**Fig 11. A schematic diagram of the distance between the adjacent neurons.**

achieved, and it was no longer practical to continue increasing the number of training steps.

**5.2.3 Attribute reduction.** The basic algorithm flow of rough-set attribute reduction is shown in Fig 12. The $C_1 - C_{11}$ attributes in the injection pump dynamometer diagram were reduced to obtain a simplified decision table.

The MATLAB data processing calculation results were:

Reduction result $c$ = [00110101100], which $C = [C_3 \ C_4 \ C_6 \ C_8 \ C_9]$

The remaining conditional properties included the following: $C_3$, $C_4$, and $C_6$ represented three Hu moment invariant features, while the two morphological eigenvalues were denoted by $C_8$ as the area eigenvalue and $C_9$ as rectangularity.

**5.2.4 LVQ neural network construction.** The structural LVQ neural network differences cause variations in its error, performance, and generalization ability. Therefore, accurate diagnostic results depend on designing the neural network to specifically identify defects in the rod pump injection-production system in the same well.

**Establishing the number of neurons**

A set of characteristic parameter data for each dynamometer diagram was reduced and yielded five columns of data ($C_3$, $C_4$, $C_6$, $C_8$, and $C_9$), representing five conditional attributes. Therefore, the number of input neurons for the LVQ neural network was set to 5, while the

**Table 7. The clustering results at different training steps.**

| Training steps | Clustering result | | | | | | | |
|---|---|---|---|---|---|---|---|---|
| 10 | 1 | 1 | 16 | 1 | 1 | 16 | 1 | 1 |
| 50 | 4 | 13 | 16 | 2 | 1 | 16 | 4 | 1 |
| 100 | 4 | 13 | 16 | 6 | 1 | 12 | 4 | 2 |

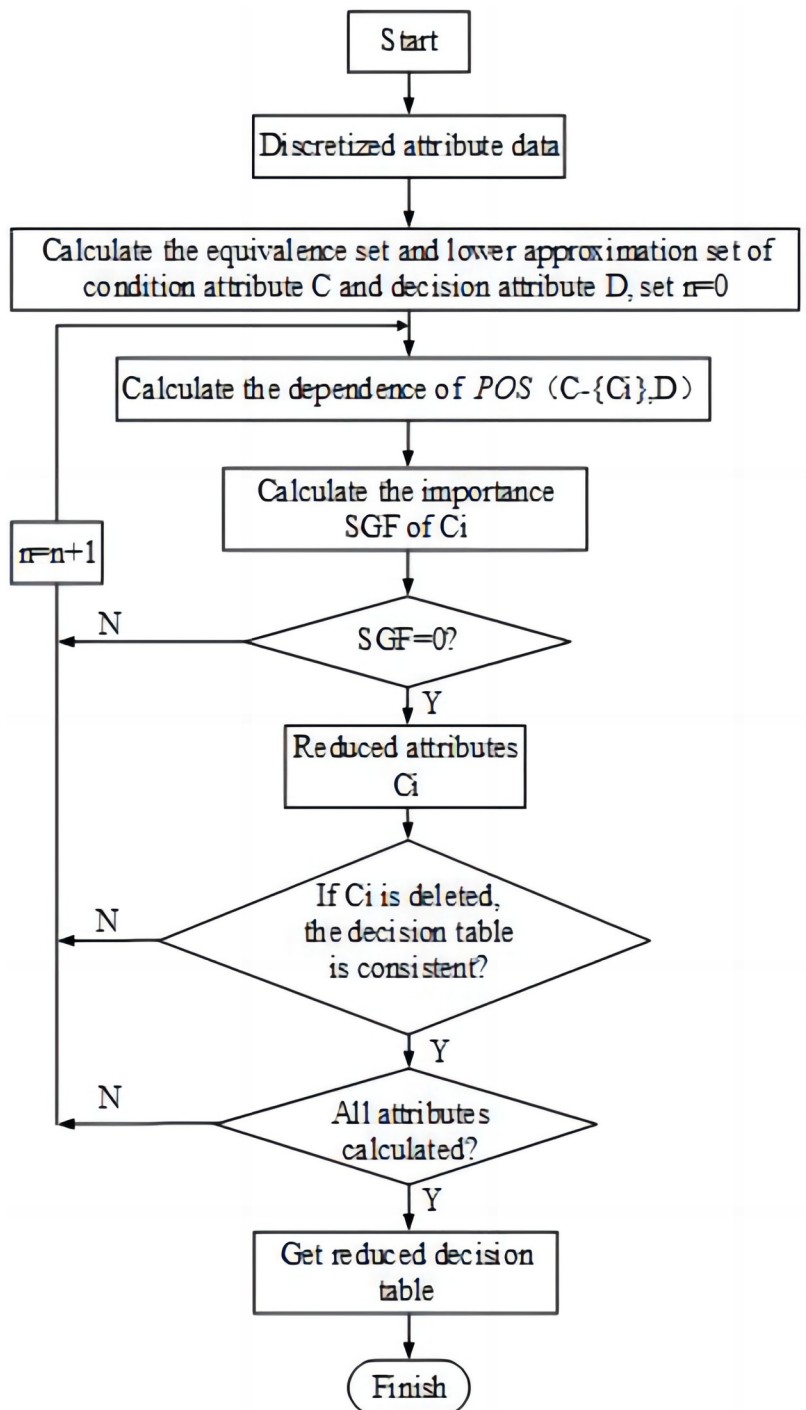

**Fig 12. The flow chart of the rough set attribute reduction algorithm.**

output neurons were set to 8, representing eight fault types. The K-fold cross-validation method was used for network training and verification, while the network error function was defined as the mean square error of the verification data. At the same training step size, the number of neurons in 20 competitive layers was tested from 1 to 20, with a corresponding

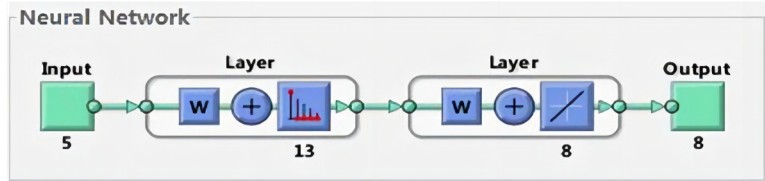

**Fig 13. The LVQ neural network structure.**

network error of:

$$[0.0092\ 0.0076\ 0.0060\ 0.0056\ 0.0055\ 0.0047\ 0.0042\ 0.0035\ 0.0030\ 0.0027$$
$$0.0024\ 0.0020\ 0.0019\ 0.0021\ 0.0025\ 0.0030\ 0.0035\ 0.0044\ 0.0048\ 0.0053]$$

The results revealed the smallest error and optimal effect at 13 neurons.

**Establishing the training parameters**

To ensure the accuracy of the classification results, the LVQ1 algorithm was used at a maximum number of iterations was set of 103, a target of 0.005, and a learning rate of 0.5.

**5.2.5 Rough set-LVQ neural network training.** After creating the neural network, the parameters could be reset and modified as necessary. Fig 13 shows a structural diagram of the constructed LVQ neural network.

After LVQ neural network training using the reduced data, its error performance and regression coordinates were checked, as shown in Figs 14 and 15, respectively.

As shown in Fig 14, the error performance of the diagnostic system decreased with an increase in the number of iterations, reaching 0.031746 at 1000 steps, it indicates that the error value is small, while the correlation coefficient between the target and actual output values was 0.92696, as illustrated in Fig 15. it shows that the predicted value is very close to the actual value.

These results indicated that the SOM neural network effectively solved the continuous attribute discretization problem in the decision table and allowed subsequent reduction processing.

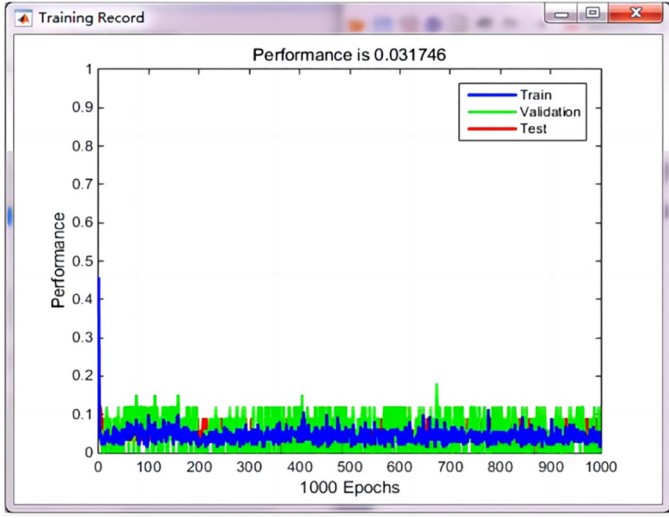

**Fig 14. The error performance graph.**

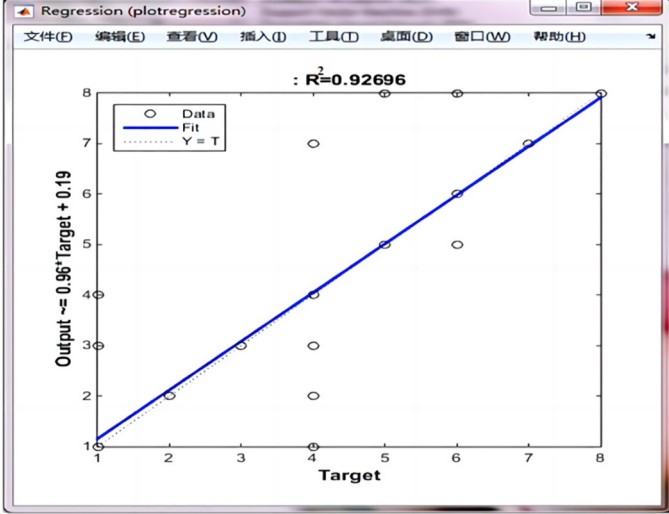

**Fig 15. The regression coordinate diagram.**

The rough set attribute reduction improved the system diagnosis efficiency and simplified the fault mode recognition LVQ network structure. Since LVQ involved back-end processing, its nonlinear mapping ability was fully utilized, guaranteeing high-precision diagnosis results.

## 5.3 Diagnostic system test

The production data information of multiple oil wells with rod pump injection and production systems were collected, and 770 typical dynamometer diagrams approved by field technicians and experts were selected for testing. The types and number of samples are shown in Table 2. The flow and diagnostic system method was used to evaluate the injection pump system failure, which was compared with the actual pump inspection results, as shown in Table 8.

**Table 8. The diagnostic system test results.**

| Number | Fault type | Number of samples | Diagnostic result | Coincidence rate |
|---|---|---|---|---|
| 1 | Upper pump valve leakage | 80 | 77 dynamometer diagrams were corrected, and 3 were diagnosed as exhibiting no faults. | 96% |
| 2 | Broken pole | 103 | 89 dynamometer diagrams were corrected, 9 were diagnosed with lower valve leakage, and 5 with continuous pumping and spraying. | 86% |
| 3 | Insufficient fluid supply | 140 | 127 dynamometer diagrams were corrected, and 13 were diagnosed with air influence. | 90% |
| 4 | Both spraying and pumping | 160 | 145 dynamometer diagrams were corrected, and 15 were diagnosed with broken poles. | 90% |
| 5 | Bump pump | 136 | All dynamometer diagrams were corrected | 100% |
| 6 | Air influence | 80 | 67 dynamometer diagrams were corrected, 11 were diagnosed with insufficient fluid supply, and 2 with lower pump valve leakage. | 83% |
| 7 | Lower pump valve leakage | 35 | 28 dynamometer diagrams were corrected, 5 were diagnosed with broken poles, and 2 with both spraying and pumping. | 80% |
| 8 | Simultaneous upper and lower pump valve leakage | 36 | 34 dynamometer diagrams were corrected, and 2 were diagnosed with upper pump valve leakage. | 94% |

The statistics show that 703 of the 770 dynamometer diagrams were correctly diagnosed, the highest accuracy is 100%, and the lowest is 80%. with a coincidence rate of 91.3%. It can be seen that even the lowest coincidence rate has good accuracy. This verified the excellent adaptability of the intelligent integrated diagnosis system for identifying faults in the rod pump injection-production system in the same well.

## 6. Conclusion

The present study establishes a fault diagnosis system for the rod pump injection and production system in the same well, based on RS-LVQ. On the premise of keeping the classification ability unchanged, the SOM neural network is used to discretize the original feature data, and the RS theory is used to reduce its attributes. After the LVQ fault diagnosis subsystem is established, the reduced decision table is input for learning and training. The results of case analysis show that: SOM neural network solves the problem of discretization of continuous attribute values in decision-making systems, and RS attribute reduction can not only improve diagnosis efficiency, but also simplify the LVQ network structure for fault pattern recognition. Strong nonlinear mapping capability ensures the accuracy of diagnosis results. Therefore, the diagnostic system can correctly and efficiently diagnose the faults of the rod pump injection-production system in the same well.

## Supporting information

**S1 File.**
(ZIP)

## Author Contributions

**Writing – original draft:** Yi fang Yin, Zunce Wang.

**Writing – review & editing:** Minzheng Jiang, Siyuan Chang.

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
