## [Decision Letter · Decision Letter 0]

22 Jun 2023

PONE-D-23-13790

Diagnosing Injection-Production System Faults in the Same Well Using the Rough Set-LVQ Neural Network

PLOS ONE

Dear Dr. Yin,

Thank you for submitting your manuscript to PLOS ONE. After careful consideration, we feel that it has merit but does not fully meet PLOS ONE’s publication criteria as it currently stands. Therefore, we invite you to submit a revised version of the manuscript that addresses the points raised during the review process.

We look forward to receiving your revised manuscript.

Kind regards,

Erman Ülker

Academic Editor

PLOS ONE

Journal Requirements:

   "This work is supported by the National Key Research and Development Program of China under grant (2022YFE0206700)."

6. Please remove your figures from within your manuscript file, leaving only the individual TIFF/EPS image files, uploaded separately. These will be automatically included in the reviewers’ PDF.

Additional Editor Comments:

The reviews of your manuscript, entitled: "Diagnosing Injection-Production System Faults in the Same Well Using the Rough Set-LVQ Neural Network", which you submitted to the PLOS ONE have now been received.

I eager to have to inform you that the referees recommended against acceptance of the paper for publication in the present situation and needs "Major Revision".

We sincerely appreciate having been given the opportunity to consider this manuscript; I hope that you will find the reviewer's remarks pertinent and helpful.

With very best wishes -

Reviewers' comments:

Reviewer's Responses to Questions

**Comments to the Author**

1. Is the manuscript technically sound, and do the data support the conclusions?

Reviewer #1: Partly

Reviewer #2: No

2. Has the statistical analysis been performed appropriately and rigorously? 

Reviewer #1: I Don't Know

Reviewer #2: No

3. Have the authors made all data underlying the findings in their manuscript fully available?

Reviewer #1: No

Reviewer #2: No

4. Is the manuscript presented in an intelligible fashion and written in standard English?

Reviewer #1: Yes

Reviewer #2: No

5. Review Comments to the Author

Reviewer #1: The manuscript entitled “Diagnosing Injection-Production System Faults in the Same Well Using the Rough Set-LVQ Neural Network” is in interest subject but some following points must be considered in revised version:

- Abbreviations such as LVQ, RS-LVQ, SOM, etc. must be introduced in the text.

- The introduction is short, some literature must be added about this object to clear it.

- In the introduction section, first sentence of the 2nd paragraph: … many studies have explored… must be sited by some studies.

- Add a summary about pumps, dynameters, etc.

- Part 2.2, reverse calculation method is not clear.

- Instead of Fig.9, add a table that introduces network parameters and structure of the used network including number hidden layer, activation functions, number of epoch, etc.

- In Fig 10, 11, 14, 15… removed gray border.

- what is the criteria of selection test data?

- Illustrate the error performances and regression plot of test and training data.

- A discussion section must be added after show results.

Reviewer #2: 1- The objectives and the rationale of the study are recommended to be clearly stated.

2- The concluding remarks of the abstract are not well-written. It's merely the repetition of the objectives and title of the manuscript. Please add quantitative findings, method limitations and justification into the abstract.

3- The innovation of using the Rough Set-LVQNeural Network is not very clear to me. It is a pretty standard model. I do not see a clear reason that this model can perform better than other methods. There are also more advanced models and why this method was selected. Why do the authors choose the model for this study?

4- The necessity & novelty of the manuscript should be presented and stressed in the "Introduction" section.

5- The application/theory/method/study reported is not in sufficient detail to allow for its replicability and/or reproducibility. Therefore, it is suggested to make it clear to show all parameters and the code.

6- The interpretation of results and study conclusions are not supported by the data. Therefore, it is recommended to deepen the discussion.

7- It is recommended to clearly emphasize the strengths of the study.

8- The limitations of the study should be stated.

9- The manuscript structure, flow or writing needs some improvements.

10- The manuscript is benefit from language editing. The English of the paper is readable; however, I would suggest the authors to have it checked preferably by a native English-speaking person to avoid any mistakes.

11- Please provide the table of hyper-parameters values of all studied algorithms.

12- Provide literature on the methods developed/applied in the "Introduction". The use of a table to demonstrate the advantage-disadvantage of these methods can be useful. Towards the end, mention the superiority & repeat the novelty of your work.

13- I would suggest that the authors review and include the following studies to improve the manuscript:

-Alakbari, Fahd Saeed, et al. "Prediction of critical total drawdown in sand production from gas wells: Machine learning approach." The Canadian Journal of Chemical Engineering (2022).

- Alakbari et al. "Prediction of bubble point pressure using artificial intelligence AI techniques." SPE middle east artificial lift conference and exhibition. OnePetro, 2016.

- Ayoub, Mohammed Abdalla, et al. "A new correlation for accurate prediction of oil formation volume factor at the bubble point pressure using Group Method of Data Handling approach." Journal of Petroleum Science and Engineering 208 (2022): 109410.

- Ayoub Mohammed, Mohammed Abdalla, et al. "Determination of the Gas–Oil Ratio below the Bubble Point Pressure Using the Adaptive Neuro-Fuzzy Inference System (ANFIS)." ACS omega 7.23 (2022): 19735-19742.

- Baarimah, Salem O., et al. "Modeling Yemeni Crude Oil Reservoir Fluid Properties Using Different Fuzzy Methods." 2022 International Conference on Data Analytics for Business and Industry (ICDABI). IEEE, 2022.

14- I noticed that the conclusion section tends to repeat the abstract and results. The conclusion paragraph should be short, impactful, and direct the reader to this research's next steps and opportunities.

6. PLOS authors have the option to publish the peer review history of their article (what does this mean?). If published, this will include your full peer review and any attached files.

Reviewer #1: No

Reviewer #2: **Yes: **Fahd Saeed Alakbari

---

## [Author Response · Author response to Decision Letter 0]

16 Aug 2023

List of responses

Title：Diagnosing Injection-Production System Faults in the Same Well Using the Rough Set-LVQ Neural Network

Manuscript ID：PONE-D-23-13790

Dear Editor and Reviewers,

On behalf of my co-authors, we are very grateful to you for giving us an opportunity to revise our manuscript. we appreciate you very much for your positive and constructive comments and suggestions on our manuscript We have studied reviewers' comments carefully and tried our best to revise our manuscript according to the comments. The following are the responses and revisions I have made in response to the reviewers' questions and suggestions on an item-by-item basis. In this revised version, changes to our manuscript were all highlighted within the document by using red-colored text. Thanks again to the hard work of the editor and reviewer!

Response to the referee's comments.

Reviewer: 1

The manuscript entitled “Diagnosing Injection-Production System Faults in the Same Well Using the Rough Set-LVQ Neural Network” is in interest subject but some following points must be considered in revised version:

We also appreciate your clear and detailed feedback and hope that the explanation has fully addressed all of your concerns. In the remainder of this letter, we discuss each of your comments individually along with our corresponding responses.

1.Abbreviations such as LVQ, RS-LVQ, SOM, etc. must be introduced in the text.

Reply: Thank you very much for this important comment.

The content “

LVQ——Learning Vector Quantization

RS——Rough Set

SOM——Self-Organizing Map

 ” has been added(Please see the revised manuscript at the first paragraph of page 1)

2.The introduction is short, some literature must be added about this object to clear it.

Reply: Thank you very much for this important comment.

The content “

 Sun et al.[10] proposed the "five-finger test" analysis method relies on experienced workers to perceive the vibration of the polished rod with their hands, and then judge the working state of the underground according to their own experience. Although easy to implement, its accuracy was low, and it was gradually replaced by other methods. Li et al.[11] proposed the surface indicator diagram method mainly uses a dynamometer to measure the suspension point diagram and matches it with a standard dynamometer diagram to obtain the downhole working status. However, this method suffers from the following problems: there are too many assumed conditions, and some of the actual measured dynamometer diagrams are too singular in shape, and it is difficult to select the closest standard diagram, thus limiting its range of application. Qiu et al[12] proposed the Downhole dynamometer method using a downhole power instrument directly downhole to measure the dynamometer diagram of the oil pump. The method eliminates many of the uncertain factors and makes the test more accurate. However, all the downhole devices need to be taken out of the ground to install the downhole dynamic instrument, and the installation cost is steep, which is not conducive to practical promotion. Gibbs and Li[13-14] proposed the Computer diagnosis method combines the mathematical model with the damping wave equation and the surface dynamometer diagram to obtain the corresponding downhole pump dynamometer diagram, and uses the dynamometer diagram to implement corresponding comparative analysis. Compared with the downhole dynamometer method, it not only saves the cost, but also improves the accuracy. However, this method still needs to rely on the skills and work experience of the operator, making it difficult to promote its use. Huang[15] proposed the “Artificial Intelligence Diagnosis” proposed by Using computer-aided diagnosis and intelligent techniques to analyze faults in the entire pumping unit, because of its advantages of simple operation and high precision, it has gradually become the current research field and has extensive development space. As shown in Table 1.

Table 1 Fault diagnosis method

 Method Advantages Shortcomings

"Five Fingers Test" Analysis Mainly rely on experienced staff to feel the vibration feeling of the polished rod in the palm, and use experience to infer the downhole working conditions Simplicity of operator Poor accuracy

Diagnosis method of ground dynamometer Use the dynamometer to measure the suspension point dynamometer map, and then compare, match and interpret it with the standard map to infer the downhole working conditions The type of failure can be more accurately defined There are many assumptions, and the shape of some measured dynamometer diagrams is too singular to select the closest standard diagram, so its application range is limited.

Downhole dynamometer diagnosis method Install the downhole power meter directly downhole to measure the dynamometer diagram of the oil well pump Many uncertain factors can be eliminated, and the test accuracy is high All downhole equipment needs to be driven out during installation, which is expensive and inconvenient for popularization and application

Computer diagnostics Establish a mathematical model based on the damped wave equation, combine the system motion law derived from the surface dynamometer diagram, solve the downhole pump dynamometer diagram, and further apply the dynamometer diagram matching method to judge the downhole working conditions Save costs, and improve Accuracy It still needs to rely on the professional skills and work experience of the operator, and there is a certain difficulty in popularizing and applying it

AI diagnostics Fault diagnosis of oil pumping system by using artificial intelligence method with the help of computer equipment Easy to operate and high precision The structural design of AI diagnosis is often determined by the designer based on experience, which has certain limitations.

” has been added(Please see the revised manuscript at the second paragraph of page 2-4)

The reference “

[10] Sun M. Fault diagnosis system for pumping wells based on neural network [D]. China University of Petroleum, 2008. 

[11] Li Z Q. Research on Lean Maintenance Technology of Rod Pumping System [D]. Wuhan University of Technology, 2009. 

[12] Qiu Z X. Research on Rod Pump Fault Diagnosis Method Based on Indicator Diagram Analysis [D]. Northeastern University, 2011. 

[13] S G Gibbs. Method of Determining Sucker Rod Pump Performance[J], US Patent, 1967, 26: 343-409. 

[14] Li K. Research on Downhole Fault Diagnosis Method of Beam Pumping Unit Based on Indicator Diagram [D]. Northeastern University, 2013. 

[15] Huang X. Intelligent Integration of Fault Diagnosis of Pumping Well Systematic research [J]. Journal of Petroleum and Natural Gas, 2007, 29(3): 156-158.

” has been added(Please see the revised manuscript at the references of page 23-24).

3.In the introduction section, first sentence of the 2nd paragraph: … many studies have explored… must be sited by some studies.

Reply: Thank you very much for this important comment.

The content “

 Sun et al.[10] proposed the "five-finger test" analysis method relies on experienced workers to perceive the vibration of the polished rod with their hands, and then judge the working state of the underground according to their own experience. Although easy to implement, its accuracy was low, and it was gradually replaced by other methods. Li et al.[11] proposed the surface indicator diagram method mainly uses a dynamometer to measure the suspension point diagram and matches it with a standard dynamometer diagram to obtain the downhole working status. However, this method suffers from the following problems: there are too many assumed conditions, and some of the actual measured dynamometer diagrams are too singular in shape, and it is difficult to select the closest standard diagram, thus limiting its range of application. Qiu et al[12] proposed the Downhole dynamometer method using a downhole power instrument directly downhole to measure the dynamometer diagram of the oil pump. The method eliminates many of the uncertain factors and makes the test more accurate. However, all the downhole devices need to be taken out of the ground to install the downhole dynamic instrument, and the installation cost is steep, which is not conducive to practical promotion. Gibbs and Li[13-14] proposed the Computer diagnosis method combines the mathematical model with the damping wave equation and the surface dynamometer diagram to obtain the corresponding downhole pump dynamometer diagram, and uses the dynamometer diagram to implement corresponding comparative analysis. Compared with the downhole dynamometer method, it not only saves the cost, but also improves the accuracy. However, this method still needs to rely on the skills and work experience of the operator, making it difficult to promote its use. Huang[15] proposed the “Artificial Intelligence Diagnosis” proposed by Using computer-aided diagnosis and intelligent techniques to analyze faults in the entire pumping unit, because of its advantages of simple operation and high precision, it has gradually become the current research field and has extensive development space. As shown in Table 1.

Table 1 Fault diagnosis method

 Method Advantages Shortcomings

"Five Fingers Test" Analysis Mainly rely on experienced staff to feel the vibration feeling of the polished rod in the palm, and use experience to infer the downhole working conditions Simplicity of operator Poor accuracy

Diagnosis method of ground dynamometer Use the dynamometer to measure the suspension point dynamometer map, and then compare, match and interpret it with the standard map to infer the downhole working conditions The type of failure can be more accurately defined There are many assumptions, and the shape of some measured dynamometer diagrams is too singular to select the closest standard diagram, so its application range is limited.

Downhole dynamometer diagnosis method Install the downhole power meter directly downhole to measure the dynamometer diagram of the oil well pump Many uncertain factors can be eliminated, and the test accuracy is high All downhole equipment needs to be driven out during installation, which is expensive and inconvenient for popularization and application

Computer diagnostics Establish a mathematical model based on the damped wave equation, combine the system motion law derived from the surface dynamometer diagram, solve the downhole pump dynamometer diagram, and further apply the dynamometer diagram matching method to judge the downhole working conditions Save costs, and improve Accuracy It still needs to rely on the professional skills and work experience of the operator, and there is a certain difficulty in popularizing and applying it

AI diagnostics Fault diagnosis of oil pumping system by using artificial intelligence method with the help of computer equipment Easy to operate and high precision The structural design of AI diagnosis is often determined by the designer based on experience, which has certain limitations.

” has been added.(Please see the revised manuscript at the second paragraph of page 2-4)

The reference “

[10] Sun M. Fault diagnosis system for pumping wells based on neural network [D]. China University of Petroleum, 2008. 

[11] Li Z Q. Research on Lean Maintenance Technology of Rod Pumping System [D]. Wuhan University of Technology, 2009. 

[12] Qiu Z X. Research on Rod Pump Fault Diagnosis Method Based on Indicator Diagram Analysis [D]. Northeastern University, 2011. 

[13] S G Gibbs. Method of Determining Sucker Rod Pump Performance[J], US Patent, 1967, 26: 343-409. 

[14] Li K. Research on Downhole Fault Diagnosis Method of Beam Pumping Unit Based on Indicator Diagram [D]. Northeastern University, 2013. 

[15] Huang X. Intelligent Integration of Fault Diagnosis of Pumping Well Systematic research [J]. Journal of Petroleum and Natural Gas, 2007, 29(3): 156-158.

” has been added.(Please see the revised manuscript at the references of page 23-24).

4.Add a summary about pumps, dynameters, etc.

Reply: Thank you very much for this important comment.

The content “

Establish and solve the static/dynamic load model of the production pump, and obtain the load-displacement function of the production pump. The load-displacement function of the suspension point derived from the suspension point indicator diagram is subtracted from the calculated load-displacement function of the production pump to obtain the load-displacement function of the injection pump so that the working condition diagnosis can be performed according to the injection pump work diagram.

” has been added.(Please see the revised manuscript at the sixth paragraph of page 6)

5.Part 2.2, reverse calculation method is not clear.

Reply: Thank you very much for this important comment.

The content “

The reverse calculation method is based on the assumption that the production pump is working normally, the fault diagnosis model of the injection-production system of the rod pump in the same well is solved by the complement grid method to obtain the production pump power diagram, and then the injection pump power diagram is obtained by further solving the collected suspension point diagram. In the normal working state of the production pump, the solution steps of the fault diagnosis model are as follows:

Fig 2. The solution steps of the fault diagnosis model

” has been added.(Please see the revised manuscript at the second paragraph of page 7)

6.Instead of Fig.9, add a table that introduces network parameters and structure of the used network including number hidden layer, activation functions, number of epoch, etc.

Reply: Thank you very much for this important comment.

The content “

Table 6 Parameters of network training

 Input layer Hidden layers Output layer

Number of neurons 5 13 8

Number of Training Epochs 100

Performance Target 0.005

Number of Validation Checks 1000

Learning Rate 0.5

” has been added(Please see the revised manuscript of page 18)

7.In Fig 10, 11, 14, 15… removed gray border.

Reply: Thank you very much for this important comment.

Fig 10, 11, 14, 15… gray border has been removed.

8.What is the criteria of selection test data?

Reply: Thank you very much for this important comment.

The criteria of selection test data are 770 dynamometer diagrams obtained through on-site testing and preliminary screening of manual fault identification.

9.Illustrate the error performances and regression plot of test and training data.

Reply: thank you very much for this important comment.

The content “

As shown in Fig 14, the error performance of the diagnostic system decreased with an increase in the number of iterations, reaching 0.031746 at 1000 steps, it indicates that the error value is small, while the correlation coefficient between the target and actual output values was 0.92696, as shown in Fig 15, it shows that the predicted value is very close to the actual value. 

” has been added(Please see the revised manuscript at the forth paragraph of page 20)

10.A discussion section must be added after show results.

Reply: Thank you very much for this important comment.

Table 6 adds a column "Accuracy"

Number Fault type Number of samples Diagnostic result Accuracy

1 Upper pump valve leakage 80 77 dynamometer diagrams

were corrected, and 3 were diagnosed as exhibiting no faults. 96%

2 Broken pole 103 89 dynamometer diagrams were corrected, 9 were diagnosed with lower valve leakage, and 5 with continuous pumping and spraying. 86%

3 Insufficient fluid supply 140 127 dynamometer diagrams were corrected, and 13 were diagnosed with air influence. 90%

4 Both spraying and pumping 160 145 dynamometer diagrams were corrected, and 15 were diagnosed with broken poles. 90%

5 Bump pump 136 All dynamometer diagrams were corrected 100%

6 Air influence 80 67 dynamometer diagrams were corrected, 11 were diagnosed with insufficient fluid supply, and 2 with lower pump valve leakage. 83%

7 Lower pump valve leakage 35 28 dynamometer diagrams were corrected, 5 were diagnosed with broken poles, and 2 with both spraying and pumping. 80%

8 Simultaneous upper and lower pump valve leakage 36 34 dynamometer diagrams were corrected, and 2 were diagnosed with upper pump valve leakage. 94%

The content “

the highest accuracy is 100%, and the lowest is 80%. With a coincidence rate of 91.3%. It can be seen that even the lowest coincidence rate has good accuracy. 

” has been added.(Please see the revised manuscript at the second paragraph of page 21)

We would like to take this opportunity to thank you for all your time involved and this great opportunity for us to improve the manuscript. We hope you will find this revised version satisfactory.

Sincerely,

Yin Yifang

Reviewer: 2

We appreciate your clear and detailed feedback and hope that the explanation has fully addressed all of your concerns. In the remainder of this letter, we discuss each of your comments individually along with our corresponding responses.

1- The objectives and the rationale of the study are recommended to be clearly stated.

Reply: Thank you very much for this important comment.

The content “

Usually, the working condition of the well can be preliminarily diagnosed through the suspended point indicator diagram, but if the working condition of the oil well is to be accurately diagnosed, it is best to use the pumping diagram. Since the downhole pump group of this system works at a depth of nearly one thousand meters underground, it is difficult to directly detect its working state parameters, and the method of using a downhole dynamic instrument is also infeasible due to factors such as inconvenient installation and high cost. A reasonable fault diagnosis model and drawing a pump power diagram are the core technologies for fault diagnosis. Through the theoretical calculation of the oil well production and the comparative analysis combined with the field survey, the statistics show that

” has been added(Please see the revised manuscript at the second paragraph of page 4)

2- The concluding remarks of the abstract are not well-written. It's merely the repetition of the objectives and title of the manuscript. Please add quantitative findings, method limitations and justification into the abstract.

Reply: Thank you very much for this important comment.

The content “

After comparing the test results with the actual working conditions, it can be seen that the rod pump injection-production diagnosis system based on RS-LVQ designed in this paper has a recognition rate of 91.3% for fault types, strong recognition ability, short diagnosis time, and A certain practicality. However, the research object of fault diagnosis in this paper is a single fault, and the actual downhole fault situation is complex, and there may be two or more fault types at the same time, which has certain limitations.

” has been added(Please see the revised manuscript at the first paragraph of page 1)

3- The innovation of using the Rough Set-LVQNeural Network is not very clear to me. It is a pretty standard model. I do not see a clear reason that this model can perform better than other methods. There are also more advanced models and why this method was selected. Why do the authors choose the model for this study?

Reply: Thank you very much for this important comment.

LVQ neural network has strong advantages in fault tolerance, self-organization ability and promotion ability. Compared with BP neural network, LVQ is slightly better in network structure, working principle and learning efficiency. However, the learning process of LVQ will also become complicated and lengthy with the increase of its network scale and the number of samples, which limits its practical promotion to a certain extent. If Rough Set(RS) is combined with it, that is, the ability of rough set to deal with uncertainty and realize attribute reduction is combined with the characteristics of LVQ, then the feature complementation can be realized and the application effect of the model can be improved.

(1) Using rough set theory as a pre-processing system: 

① Reduce the spatial dimension of input information, simplify the network structure of LVQ; 

② Reduce the number of training samples and shorten the training time. 

(2) Using LVQ as the back-end processing system: 

① Give full play to its strong nonlinear mapping ability to ensure the accuracy of diagnosis results; ② It can better fault tolerance and anti-interference, and will not be interrupted by various reasons.

At the same time, under the premise of keeping the classification ability unchanged, the SOM neural network is used to discretize the original feature data, and the RS theory can be used to reduce its attributes. After the LVQ fault diagnosis subsystem is established, the reduction decision table is input. Learn to train. The results of case analysis show that: SOM neural network solves the problem of discretization of continuous attribute values in decision-making systems, and RS attribute reduction can not only improve diagnosis efficiency, but also simplify the LVQ network structure for fault pattern recognition. Strong nonlinear mapping capability ensures the accuracy of diagnosis results. So RS-LVQ is chosen as the fault diagnosis system model in this paper.

4-The necessity & novelty of the manuscript should be presented and stressed in the "Introduction" section.

Reply: Thank you very much for this important comment.

The content “

 According to the commonly used method, using the suspension point dynamometer diagram for fault diagnosis, it is impossible to accurately identify and distinguish the fault types of the injection pump and the recovery pump, Therefore, based on an actual site scenario, this paper proposes a reverse calculation method that considers the production pump as a supplementary boundary condition and uses RS-LVQ to establish a downhole technique for diagnosing faults in the rod pump injection and production system in the same well. Therefore, defects are rapidly and accurately identified, allowing targeted measures to be implemented to address the issue[11-13], making up for the shortcomings of traditional methods.” has been added(Please see the revised manuscript at the second paragraph of page 4)

5-The application/theory/method/study reported is not in sufficient detail to allow for its replicability and/or reproducibility. Therefore, it is suggested to make it clear to show all parameters and the code.

Reply: Thank you very much for this important comment.

The model has been experimented with more than 700 dynamometer diagrams, and it has been verified that the model has good repeatability. And we did our best to get permission to upload 770 indicator diagrams. However, the participating projects involve confidentiality, so we cannot provide the corresponding code, please forgive us.

6-The interpretation of results and study conclusions are not supported by the data. Therefore, it is recommended to deepen the discussion.

Reply: Thank you very much for this important comment.

The content “

The present study establishes a fault diagnosis system for the rod pump injection and production system in the same well, based on RS-LVQ. On the premise of keeping the classification ability unchanged, the SOM neural network is used to discretize the original feature data, and the RS theory is used to reduce its attributes. After the LVQ fault diagnosis subsystem is established, the reduced decision table is input for learning and training. The results of case analysis show that: SOM neural network solves the problem of discretization of continuous attribute values in decision-making systems, and RS attribute reduction can not only improve diagnosis efficiency, but also simplify the LVQ network structure for fault pattern recognition. Strong nonlinear mapping capability ensures the accuracy of diagnosis results. Therefore, the diagnostic system can correctly and efficiently diagnose the faults of the rod pump injection-production system in the same well. 

” has been added(Please see the revised manuscript at the last paragraph of page 21-22)

7-It is recommended to clearly emphasize the strengths of the study.

Reply: Thank you very much for this important comment.

Aiming at the unconventional structure of rod pump injection-production system in the same well, a reverse calculation method is proposed, and its fault diagnosis model is established and solved, and then the seven invariant moments of the injection pump power diagram are extracted by using the invariant moment characteristic method. Based on RS-LVQ, a downhole fault diagnosis system for rod pump injection and production system in the same well was established. On the premise of keeping the classification ability unchanged, the SOM neural network is used to discretize the original feature data, and the RS theory is used to reduce its attributes. After the LVQ fault diagnosis subsystem is established, the reduced decision table is input for learning and training. The results of case analysis show that: SOM neural network solves the problem of discretization of continuous attribute values in decision-making systems, and RS attribute reduction can not only improve diagnosis efficiency, but also simplify the LVQ network structure for fault pattern recognition. Strong nonlinear mapping capability ensures the accuracy of diagnosis results. Therefore, the diagnostic system can correctly and efficiently diagnose the faults of the rod pump injection-production system in the same well.

8-The limitations of the study should be stated.

Reply: Thank you very much for this important comment.

The content “

However, the research object of fault diagnosis in this paper is a single fault, and the actual downhole fault situation is complex, and there may be two or more fault types at the same time, which has certain limitations.

” has been added(Please see the revised manuscript at the last paragraph of page 1)

9-The manuscript structure, flow or writing needs some improvements.

Reply: Thank you very much for this important comment.

We have tried our best to polish the language in the revised manuscript.

10-The manuscript is benefit from language editing. The English of the paper is readable; however, I would suggest the authors to have it checked preferably by a native English-speaking person to avoid any mistakes.

Reply: Thank you very much for this important comment.

However, we do invite a friend of us who is a native English speaker to help polish our article. And we hope the revised manuscript could be acceptable for you.

11-Please provide the table of hyper-parameters values of all studied algorithms.

Reply: Thank you very much for this important comment.

Participating projects and data involve confidentiality agreements, so we cannot provide the corresponding data, please forgive us.

12-Provide literature on the methods developed/applied in the "Introduction". The use of a table to demonstrate the advantage-disadvantage of these methods can be useful. Towards the end, mention the superiority & repeat the novelty of your work.

Reply: Thank you very much for this important comment.

The content “

 Method Advantages Shortcomings

"Five Fingers Test" Analysis Mainly rely on experienced staff to feel the vibration feeling of the polished rod in the palm, and use experience to infer the downhole working conditions Simplicity of operator Poor accuracy

Diagnosis method of ground dynamometer Use the dynamometer to measure the suspension point dynamometer map, and then compare, match and interpret it with the standard map to infer the downhole working conditions The type of failure can be more accurately defined There are many assumptions, and the shape of some measured dynamometer diagrams is too singular to select the closest standard diagram, so its application range is limited.

Downhole dynamometer diagnosis method Install the downhole power meter directly downhole to measure the dynamometer diagram of the oil well pump Many uncertain factors can be eliminated, and the test accuracy is high All downhole equipment needs to be driven out during installation, which is expensive and inconvenient for popularization and application

Computer diagnostics Establish a mathematical model based on the damped wave equation, combine the system motion law derived from the surface dynamometer diagram, solve the downhole pump dynamometer diagram, and further apply the dynamometer diagram matching method to judge the downhole working conditions Save costs, and improve Accuracy It still needs to rely on the professional skills and work experience of the operator, and there is a certain difficulty in popularizing and applying it

AI diagnostics Fault diagnosis of oil pumping system by using artificial intelligence method with the help of computer equipment Easy to operate and high precision The structural design of AI diagnosis is often determined by the designer based on experience, which has certain limitations.

According to the commonly used method, using the suspension point dynamometer diagram for fault diagnosis, it is impossible to accurately identify and distinguish the fault types of the injection pump and the recovery pump, Therefore, based on an actual site scenario, this paper proposes a reverse calculation method that considers the production pump as a supplementary boundary condition and uses RS-LVQ to establish a downhole technique for diagnosing faults in the rod pump injection and production system in the same well. Therefore, defects are rapidly and accurately identified, allowing targeted measures to be implemented to address the issue[11-13], making up for the shortcomings of traditional methods.

” has been added(Please see the revised manuscript at the first paragraph of page 3-4)

13- I would suggest that the authors review and include the following studies to improve the manuscript:

 Alakbari, Fahd Saeed, et al. "Prediction of critical total drawdown in sand production from gas wells: Machine learning approach." The Canadian Journal of Chemical Engineering (2022).

 Alakbari et al. "Prediction of bubble point pressure using artificial intelligence AI techniques." SPE middle east artificial lift conference and exhibition. OnePetro, 2016.

 Ayoub, Mohammed Abdalla, et al. "A new correlation for accurate prediction of oil formation volume factor at the bubble point pressure using Group Method of Data Handling approach." Journal of Petroleum Science and Engineering 208 (2022): 109410.

 Ayoub Mohammed, Mohammed Abdalla, et al. "Determination of the Gas–Oil Ratio below the Bubble Point Pressure Using the Adaptive Neuro-Fuzzy Inference System (ANFIS)." ACS omega 7.23 (2022): 19735-19742.

 Baarimah, Salem O., et al. "Modeling Yemeni Crude Oil Reservoir Fluid Properties Using Different Fuzzy Methods." 2022 International Conference on Data Analytics for Business and Industry (ICDABI). IEEE, 2022.

Reply: Thank you very much for this important comment.

[6] Alakbari, Fahd Saeed, et al. "Prediction of critical total drawdown in sand production from gas wells: Machine learning approach." The Canadian Journal of Chemical Engineering (2022).

[7] Alakbari et al. "Prediction of bubble point pressure using artificial intelligence AI techniques." SPE middle east artificial lift conference and exhibition. OnePetro, 2016.

[8] Ayoub Mohammed, Mohammed Abdalla, et al. "Determination of the Gas–Oil Ratio below the Bubble Point Pressure Using the Adaptive Neuro-Fuzzy Inference System (ANFIS)." ACS omega 7.23 (2022): 19735-19742.

[9] Baarimah, Salem O., et al. "Modeling Yemeni Crude Oil Reservoir Fluid Properties Using Different Fuzzy Methods." 2022 International Conference on Data Analytics for Business and Industry (ICDABI). IEEE, 2022.

[22] Ayoub, Mohammed Abdalla, et al. "A new correlation for accurate prediction of oil formation volume factor at the bubble point pressure using Group Method of Data Handling approach." Journal of Petroleum Science and Engineering 208 (2022): 109410.

We have cited related literature in the proper place of the revised manuscript.

14-I noticed that the conclusion section tends to repeat the abstract and results. The conclusion paragraph should be short, impactful, and direct the reader to this research's next steps and opportunities.

Reply: Thank you very much for this important comment.

The present study establishes a fault diagnosis system for the rod pump injection and production system in the same well, based on RS-LVQ. On the premise of keeping the classification ability unchanged, the SOM neural network is used to discretize the original feature data, and the RS theory is used to reduce its attributes. After the LVQ fault diagnosis subsystem is established, the reduced decision table is input for learning and training. The results of case analysis show that: SOM neural network solves the problem of discretization of continuous attribute values in decision-making systems, and RS attribute reduction can not only improve diagnosis efficiency, but also simplify the LVQ network structure for fault pattern recognition. Strong nonlinear mapping capability ensures the accuracy of diagnosis results. Therefore, the diagnostic system can correctly and efficiently diagnose the faults of the rod pump injection-production system in the same well. The details are as follows:

” has been added(Please see the revised manuscript at the last paragraph of page 21-22)

We would like to take this opportunity to thank you for all your time involved and this great opportunity for us to improve the manuscript. We hope you will find this revised version satisfactory.

Sincerely,

Yin Yifang

Funding Statement :"This work is supported by the National Key Research and Development Program of China under grant (2022YFE0206700)."

---

## [Decision Letter · Decision Letter 1]

29 Aug 2023

Diagnosing injection-production system faults in the same well using the RS-LVQ neural network

PONE-D-23-13790R1

Dear Dr.Yin,

We’re pleased to inform you that your manuscript has been judged scientifically suitable for publication and will be formally accepted for publication once it meets all outstanding technical requirements.

Kind regards,

Erman Ülker

Academic Editor

PLOS ONE

---

## [Editor Report · Acceptance letter]

14 Nov 2023

PONE-D-23-13790R1 

Diagnosing Injection-Production System Faults in the Same Well Using the Rough Set-LVQ Neural Network 

Dear Dr. Yin:

I'm pleased to inform you that your manuscript has been deemed suitable for publication in PLOS ONE. Congratulations! Your manuscript is now with our production department. 

Kind regards, 

on behalf of

Dr. Erman Ülker 

Academic Editor

PLOS ONE